# Simple, Good, Fast: Self-Supervised World Models Free of Baggage

**Jan Robine**[1,2] **Marc Höftmann**[1,2] **& Stefan Harmeling**[1,2]
[1]TU Dortmund, [2]Lamarr Institute for Machine Learning and Artificial Intelligence
{jan.robine,marc.hoeftmann,stefan.harmeling}@tu-dortmund.de

## Abstract

What are the essential components of world models? How far do we get with world models that are not employing RNNs, transformers, discrete representations, and image reconstructions? This paper introduces SGF, a Simple, Good, and Fast world model that uses self-supervised representation learning, captures short-time dependencies through frame and action stacking, and enhances robustness against model errors through data augmentation. We extensively discuss SGF's connections to established world models, evaluate the building blocks in ablation studies, and demonstrate good performance through quantitative comparisons on the Atari 100k benchmark. The code is available at https://github.com/jrobine/sgf.

## 1 Introduction

Deep reinforcement learning has demonstrated remarkable success in solving challenging decision-making problems (Mnih et al., 2015; Schulman et al., 2017; Mnih et al., 2016; Hessel et al., 2018; Badia et al., 2020; Schrittwieser et al., 2020; Kapturowski et al., 2023; Hafner et al., 2023). Despite these achievements, the primary challenge remains sample efficiency, i.e., the amount of data required to learn effective behaviors. Recent works have addressed this challenge by improving architectures and hyperparameters (van Hasselt et al., 2019; Schwarzer et al., 2023), pretraining and fine-tuning (Schwarzer et al., 2021b), applying data augmentation (Yarats et al., 2021; Laskin et al., 2020a), incorporating ideas from self-supervised representation learning (Laskin et al., 2020b; Schwarzer et al., 2021a;b; 2023), or learning a model of the environment (Kaiser et al., 2020; Ye et al., 2021; Robine et al., 2023; Micheli et al., 2023; Hafner et al., 2020; 2021; 2023).

Several approaches have been proposed in the literature to leverage a model of the environment. Improving the representations for the model-free agent can be achieved by learning a (latent) transition model (Lee et al., 2020; Schwarzer et al., 2021a), a reward model, or both (Gelada et al., 2019; Zhang et al., 2021), where the model serves as an auxiliary task. Aside from that, a *world model*, i.e., a deep generative model of the environment, can be learned. World models find applications in learning in imagination, where a model-free algorithm is applied to sequences generated by the world model (Sutton, 1991; Ha & Schmidhuber, 2018; Kaiser et al., 2020; Hafner et al., 2020; 2021; 2023; Micheli et al., 2023; Robine et al., 2023), and in decision-time planning, where the model is used for lookahead

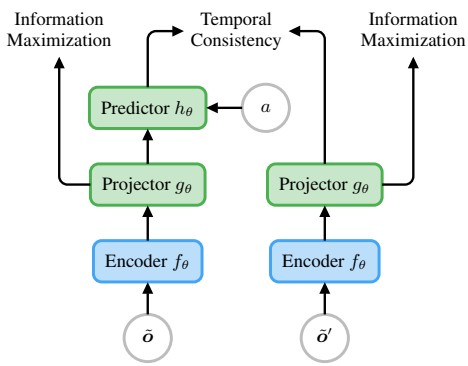

Figure 1: Our world model learns representations that are both temporally consistent and maximize the information about the observations.

search during action selection (Watter et al., 2015; Banijamali et al., 2018; Chua et al., 2018; Hafner et al., 2019). Another line of work performs decision-time planning without a full world model, relying on *value equivalence*. In this paradigm, trajectories of the model are required to yield the same cumulative rewards as those in the real environment, regardless of whether the produced hidden states correspond to any real environment states or not (Tamar et al., 2016; Silver et al., 2017; Oh et al., 2017; Schrittwieser et al., 2020; Ye et al., 2021; Hansen et al., 2022).

In computer vision, self-supervised learning of image representations has made significant progress in recent years (Chen et al., 2020; He et al., 2020; Grill et al., 2020; Caron et al., 2020; Chen & He, 2021; Zbontar et al., 2021; Caron et al., 2021; Bardes et al., 2022). Many approaches are based on Siamese neural networks (Bromley et al., 1993) and can be categorized into contrastive and non-contrastive methods. Contrastive methods (Chen et al., 2020; He et al., 2020) aim to learn representations that are similar for different views (e.g. image augmentations) of the same image but dissimilar for different images to prevent the collapse of representations. Non-contrastive methods do not rely on negative samples. Instead, they prevent representation collapse by the design of the architecture (Grill et al., 2020; Chen & He, 2021) or by regularization of the representations (Zbontar et al., 2021; Bardes et al., 2022).

**Contributions:**  In this work, we explore simplifying world models while maintaining good performance. Many world models rely on RNNs or transformers to capture long-term dependencies, introducing computational complexity and instability. We focus on problems where short-term dependencies might be sufficient, *avoiding sequence models* and instead leveraging self-supervised learning, data augmentation, and stacking. Our contributions are as follows:

- We present SGF, a world model based on a representation learning framework inspired by VICReg (Bardes et al., 2022). While restricting ourselves to simple world models (w/o RNNs and transformers) and choosing simple ingredients, we still have the essential properties of effective world models: *maximum information* and *temporal consistency* (Section 2.3).
- We reduce the complexity of world models in terms of both methodology and implementation: SGF does not require image reconstructions or discretization of representations. Besides avoiding sequence models, such as recurrent neural networks or transformers, we also do not use probabilistic predictions for deterministic environments. Instead, we employ simple techniques from model-free reinforcement learning, in particular data augmentation and stacking, which already have been successfully used before (Section 2.2).
- We conduct several ablations and thoroughly discuss the similarities and differences between SGF and other world models that learn in imagination (Sections 4.2 and 5.2).
- We demonstrate that our design choices lead to shorter training times compared to other world models, while achieving good performance on the Atari 100k benchmark (Section 5.1).

## 2 SIMPLE, GOOD, AND FAST WORLD MODELS

While being increasingly powerful, model-based approaches have simultaneously grown in complexity and consist of more and more components that need to be adjusted to each other (Hafner et al., 2020; 2021; 2023; Robine et al., 2023; Micheli et al., 2023). After introducing our notation, we aim to distill minimal ingredients and objectives for world models that are easy to implement and computationally efficient.

### 2.1 PRELIMINARIES AND NOTATION

We formalize the environment in terms of a partially observable Markov decision process (POMDP) with discrete time steps, rewards $r \in \mathbb{R}$, image observations $\boldsymbol{o} \in \mathbb{R}^{C \times H \times W}$, and actions $a \in \mathcal{A}$, which are either discrete or continuous. Transitions within the environment are described by a tuple $(\boldsymbol{o}, \boldsymbol{a}, \boldsymbol{o}', r, e)$, where $\boldsymbol{o}$ is the current observation, $\boldsymbol{o}'$ is the next observation, and $e \in \{0, 1\}$ indicates terminal states.

In its simplest form, a world model is a generative model of the dynamics $p(\boldsymbol{o}', r, e \mid \boldsymbol{o}, \boldsymbol{a})$ of a POMDP. Given a policy $\boldsymbol{a} = \pi(\boldsymbol{o})$, iterative sampling from the world model generates trajectories without further real environment interactions. These trajectories can be used for learning behaviors in imagination (Ha & Schmidhuber, 2018; Hafner et al., 2020), e.g., via model-free RL.

To increase efficiency, world models should operate in a low-dimensional representation space (commonly known as latent space) as opposed to the high-dimensional observation space (Ha & Schmidhuber, 2018). For this we need two components: a *representation model* that maps image observations $\boldsymbol{o}$ onto representations $\boldsymbol{y}$, and a *dynamics model* that predicts the latent dynamics $p(\boldsymbol{y}', r, e \mid \boldsymbol{y}, \boldsymbol{a})$. Usually, the policy also operates in the low-dimensional space, i.e., $\boldsymbol{a} = \pi(\boldsymbol{y})$, enabling behavior learning with high computational efficiency (Ha & Schmidhuber, 2018).

A common assumption is the conditional independence of $r$ and $e$ given $\boldsymbol{y}$, $\boldsymbol{a}$, and $\boldsymbol{y}'$, which is also employed by previous world models (e.g., Micheli et al., 2023; Hafner et al., 2023). This leads to the following factorization of the latent dynamics

$$p(\boldsymbol{y}', r, e \mid \boldsymbol{y}, \boldsymbol{a}) = p(\boldsymbol{y}' \mid \boldsymbol{y}, \boldsymbol{a})\, p(r \mid \boldsymbol{y}, \boldsymbol{a}, \boldsymbol{y}')\, p(e \mid \boldsymbol{y}, \boldsymbol{a}, \boldsymbol{y}'), \tag{1}$$

which consists of three conditional distributions: the *transition distribution* $p(\boldsymbol{y}' \mid \boldsymbol{y}, \boldsymbol{a})$, which is only conditioned on $\boldsymbol{y}$ and $\boldsymbol{a}$, the *reward distribution* $p(r \mid \boldsymbol{y}, \boldsymbol{a}, \boldsymbol{y}')$ and the *terminal distribution* $p(e \mid \boldsymbol{y}, \boldsymbol{a}, \boldsymbol{y}')$, both further conditioned on the next representation $\boldsymbol{y}'$. This is a natural choice for many POMDPs, where most of the complexity of the dynamics is captured by the transition distribution. This allows for learning three separate models rather than modeling the joint distribution.

## 2.2 Ingredients Leading to Simplicity

**Stacking instead of memory.** Model-free methods often assume that the observations of an POMDP approximately satisfy the Markov property. This means that the next observation $\boldsymbol{o}'$ (and consequently $\boldsymbol{y}'$) is independent of the preceding history of transitions given the current observation $\boldsymbol{o}$ and action $\boldsymbol{a}$. However, previous works on world models consider *non*-Markovian observations, which might exhibit long-term dependencies. This is approached by adding a notion of *memory* to the dynamics model, which can be realized by introducing recurrent states (RNN-based, Ha & Schmidhuber, 2018; Hafner et al., 2020) or by directly conditioning on the history of transitions (attention-based Robine et al., 2023; Micheli et al., 2023). However, the memory is typically a big computational burden and makes the model more complicated. We are asking whether we can omit the memory to obtain a much faster world model.

To capture short-time dependencies with minimal computational overhead, we suggest to simply use *frame and action stacking*. Stacking the $m$ most recent *frames* alleviates the problem of partial observability, e.g., by capturing the velocity of objects in the scene. This is a well-known preprocessing technique in model-free reinforcement learning (Mnih et al., 2015), and has already been applied to world models by Robine et al. (2023). Additionally, stacking the most $m$ recent *actions* can be beneficial, considering potential delays in the effects of actions. In Section 4.2, we show significant improvements of our world model through action stacking while being computationally cheap.

**Augmentations instead of stochasticity.** In deterministic POMDPs, executing a specific action in a specific state consistently yields the same outcomes $\boldsymbol{o}'$, $r$, and $e$. However, prior world models are stochastic even in deterministic POMDPs (see Section 5.2). Ha & Schmidhuber (2018) argue that, due to model errors, behaviors learned in imagination may perform poorly in the real environment. They propose that stochastic predictions can reduce the exploitability of an imperfect world model.

Similarly, we introduce stochasticity through *data augmentation*, as demonstrated in previous model-free algorithms (Yarats et al., 2021; Laskin et al., 2020a). We investigate whether this helps to improve the robustness of our world model against model errors. In Section 4.2, we demonstrate that data augmentation significantly improves the performance of our world model.

## 2.3 Essential Properties of Representations

Building meaningful representations of observations is crucial for dynamics modeling and behavior learning. In this work, we argue that representations should possess two key properties: (1) the information about the observations should be maximized, (2) they should be temporally consistent, i.e., representations of two successive observations should be similar. We describe these properties in more detail below. Both properties are already present in other world models (see Section 5.2), however, in this work we try to implement them in a most simple and efficient manner.

**Maximizing information.** In latent world models, the representations are used for downstream behavior learning, so any information not encoded in these representations is not accessible to the agent. Therefore, extracting maximum information from observations is necessary for learning optimal behaviors in latent space. This is often realized by reconstructing the input observations (i.e., autoencoder-style), however, this can also be realized by self-supervised objectives, as we show in Section 3.

**Temporal consistency.** Temporal consistency can be motivated by *predictive coding*, where the future or missing information is predicted. Predictive coding has been applied in information theory for data compression (Elias, 1955), and more recently in representation learning (Oord et al., 2018; Hénaff et al., 2019). Also, the neuroscience literature suggests that the human brain learns internal representations of incoming sensory signals by minimizing prediction errors subject to particular constraints on the representations, in spatial and temporal domains (Rao & Ballard, 1999; Hosoya et al., 2005; Huang & Rao, 2011). Furthermore, Hénaff et al. (2019) proposed the *temporal straightening hypothesis* which suggests that inside the human brain visual inputs are transformed to follow straighter temporal trajectories, in order to make the stream of visual inputs more predictable. These advantages of *temporal consistency* for biological agents translate to advantages for agents in reinforcement learning that are based on latent world models:

- *Simpler dynamics prediction:* as successive observations are close in representation space, the dynamics model often only needs to predict minor changes and the danger of sudden jumps in representation space is reduced.

- *Improved behavior learning:* both, the policy and value function benefit from temporally similar representations. This concept can be loosely connected to *bisimulation metrics*, where "behaviorally similar" states are grouped together (Zhang et al., 2021). Further details are available in Appendix B.1.

## 3 BUILDING BLOCKS FOR SGF

Having identified core ingredients, we now describe the construction of a simple, fast, and good world model. We will begin by outlining the representation model of SGF, followed by an description of the dynamics model, implementational details, and our evaluation protocol. For representation learning we draw inspiration from VICReg (Bardes et al., 2022). Further connections to other self-supervised methods are discussed in Appendix B.2.

### 3.1 LEARNING A WORLD MODEL

**Representation learning.** Given a POMDP transition $(\boldsymbol{o}, \boldsymbol{a}, \boldsymbol{o}', r, e)$, we apply random transformations $t, t' \sim \mathcal{T}$ from a set $\mathcal{T}$ of image augmentations to obtain augmented observations $\tilde{\boldsymbol{o}} = t(\boldsymbol{o})$ and $\tilde{\boldsymbol{o}}' = t'(\boldsymbol{o}')$. An encoder $f_\theta$ computes representations $\tilde{\boldsymbol{y}} = f_\theta(\tilde{\boldsymbol{o}})$ and $\tilde{\boldsymbol{y}}' = f_\theta(\tilde{\boldsymbol{o}}')$ with $\tilde{\boldsymbol{y}}, \tilde{\boldsymbol{y}}' \in \mathbb{R}^d$. A projector network $g_\theta$ computes embeddings $\tilde{\boldsymbol{z}} = g_\theta(\tilde{\boldsymbol{y}})$ and $\tilde{\boldsymbol{z}}' = g_\theta(\tilde{\boldsymbol{y}}')$ with $\tilde{\boldsymbol{z}}, \tilde{\boldsymbol{z}}' \in \mathbb{R}^D$. An action-conditioned predictor network $h_\theta$ predicts the next embedding $\hat{\boldsymbol{z}}' = h_\theta(\tilde{\boldsymbol{z}}, a)$. An illustration can be seen in Figure 1.

To achieve temporal consistency, we minimize the mean squared error between $\hat{z}'$ and $\tilde{z}'$. To maximize the information content, the embeddings are regularized using the variance and covariance regularization terms proposed by Bardes et al. (2022). The total representation loss is summarized by

$$\mathcal{L}_{\text{Repr.}}(\theta) = \mathbb{E}_\tau \Big[ \underbrace{\tfrac{\eta}{D} \| h_\theta(\tilde{\boldsymbol{z}}, \mathbf{a}) - \tilde{\boldsymbol{z}}' \|_2^2}_{\text{Temporal Consistency}} + \underbrace{\text{VC}(\tilde{\mathbf{Z}}) + \text{VC}(\tilde{\mathbf{Z}}')}_{\text{Information Maximization}} \Big], \tag{2}$$

where $\tau$ is a batch of transitions from a replay buffer, $\tilde{\mathbf{Z}}$ and $\tilde{\mathbf{Z}}'$ are batches of embeddings, $\eta > 0$ controls the strength of the consistency loss, and VC (variance and covariance) is defined as

$$\text{VC}(\mathbf{Z}) = \frac{1}{D} \sum_{j=1}^{D} \Big[ \underbrace{\rho \max\Big(0, 1 - \sqrt{\text{Cov}(\mathbf{Z})_{j,j} + \varepsilon}\Big)}_{\text{Variance Regularization}} + \underbrace{\nu \sum_{k \neq j} \text{Cov}(\mathbf{Z})_{j,k}^2}_{\text{Covariance Regularization}} \Big], \tag{3}$$

where $D$ is the dimensionality of the embeddings, $\rho, \nu > 0$ control the strength of variance and covariance regularization terms, respectively, and $\varepsilon = 1 \times 10^{-4}$ prevents numerical instabilities. The goal of the VC terms is to maximize information content and to prevent representation collapse. Variance regularization keeps the standard deviation of each embedding feature across the batch above 1 using a hinge loss. Covariance regularization decorrelates the embedding features by attracting their covariances towards zero (Bardes et al., 2022).

**Dynamics learning.** We build a simple dynamics model and rely on the capabilities of temporally consistent representations. Based on the dynamics factorization, we learn a transition distribution $p_\theta(\boldsymbol{y}' \mid \boldsymbol{y}, \boldsymbol{a})$, a reward distribution $p_\theta(r \mid \boldsymbol{y}, \boldsymbol{a}, \boldsymbol{y}')$, and a terminal distribution $p_\theta(e \mid \boldsymbol{y}, \boldsymbol{a}, \boldsymbol{y}')$. In this work, we focus on deterministic prediction, i.e., we simply calculate the means for transitions and rewards, and the mode for terminals (more details on this in Appendix F). Maximum likelihood estimation leads to the total dynamics loss

$$\mathcal{L}_{\text{Dyn.}}(\theta) = \mathbb{E}_\tau \Big[ \underbrace{- \log p_\theta(\text{sg}(\mathbf{y}') \mid \text{sg}(\mathbf{y}), \mathbf{a})}_{\text{Transition Distribution}} \underbrace{- \log p_\theta(\text{r} \mid \tilde{\mathbf{y}}, \text{a}, \tilde{\mathbf{y}}')}_{\text{Reward Distribution}} \underbrace{- \log p_\theta(\text{e} \mid \tilde{\mathbf{y}}, \text{a}, \tilde{\mathbf{y}}')}_{\text{Terminal Distribution}} \Big], \quad (4)$$

where $\text{sg}(\cdot)$ denotes the stop-gradient operator, meaning that the representations are not influenced by the loss of the transition distribution. This is because the transition distribution has moving targets, given that $\boldsymbol{y}'$ originates from the representation model, which changes during training. The rewards and terminals provide stable signals from the POMDP. Note that we learn the transitions with non-augmented observations, i.e., $\boldsymbol{y} = f_\theta(\boldsymbol{o})$ and $\boldsymbol{y}' = f_\theta(\boldsymbol{o}')$.

### 3.2 Learning Behaviors in Imagination

The representations $\boldsymbol{y}$ serve as inputs to the policy $\pi_\phi(\boldsymbol{a} \mid \boldsymbol{y})$. Through iterative dynamics prediction, batches of representations, actions, rewards, and terminals are generated and used to train the policy. The policy learns to maximize the expected return by performing approximate gradient ascent with the policy gradient (Sutton et al., 1999). To reduce the variance of the gradient estimates, we employ a learned value function $v_\phi(\boldsymbol{y})$ as a baseline, resulting in an advantage actor-critic approach (Mnih et al., 2016); the details are explained in Appendix F. The pseudocode outlining our world model and policy training procedure is presented in Algorithm 1.

### 3.3 Evaluation Protocol

We evaluate our world model on the Atari 100k benchmark, which was first proposed by Kaiser et al. (2020) and has been used to evaluate many sample-efficient reinforcement learning methods (Laskin et al., 2020b; Yarats et al., 2021; Schwarzer et al., 2021a; 2023; Micheli et al., 2023; Hafner et al., 2023). It includes a subset of 26 Atari games from the Arcade Learning Environment (Bellemare et al., 2013) and is limited to 400k environment steps, which amounts to 100k steps after frame skipping or 2 hours of human gameplay. Note that all games are deterministic (Machado et al., 2018).

We perform 10 runs per game and for each run we compute the average score over 100 episodes at the end of training. We follow Micheli et al. (2023) by selecting a random action with $1\%$ probability inside the environment and using a sampling temperature of $0.5$ for the policy during evaluation. We also adapt their special handling of Freeway and use a sampling temperature of $0.01$ for the policy.

## 4 Empirical Study

Behavior learning depends entirely on the quality of the world model. To assess our world model, we first analyze it qualitatively and then show how getting rid of individual design choices results in a degradation of performance. Further analysis and ablations are presented in Appendices C and E.

### 4.1 Inspecting the World Model

To assess whether the learned representations contain relevant information and do not collapse, we train a separate decoder for analysis without affecting the world model's gradients. This allows us to visually interpret the imagined sequences of the world model. In Figure 2 we depict three exemplary sequences, demonstrating that the learned representations contain useful information without relying on image reconstructions. We can also see that the dynamics model can work for long sequences (30 time steps) without accumulating notable model errors, although being only a feedforward model.

Furthermore, we study the temporal consistency of the learned representations. For that, we generate an episode in Pong following a policy trained with our approach. We then encode the observations and compute two-dimensional t-SNE embeddings (Van der Maaten & Hinton, 2008) of the representations. The result can be seen in Figure 3, and it suggests that subsequent observations have similar

 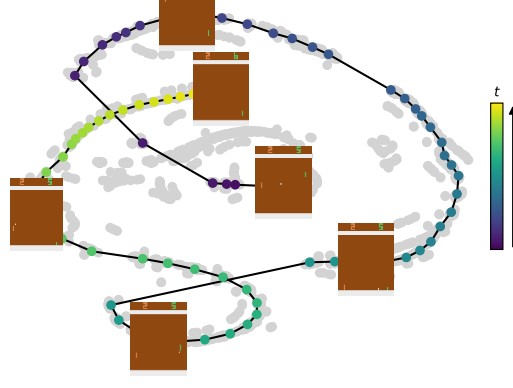

Figure 2: Illustration of imagined sequences of length 30. Each frame in the frame stack is converted to grayscale, and pixel changes are visualized in red, green, and blue. From top to bottom: Kung Fu Master, Ms Pacman, and Seaquest.

Figure 3: Two-dimensional t-SNE embeddings of the learned representations obtained by playing an episode in Pong. We highlight a subsequence of the episode, which starts with a new ball and stops after a point is scored.

representations and temporally consistency is successfully employed. In Figure 7 we compare the learned embeddings when disabling temporal consistency.

## 4.2 ABLATING THE WORLD MODEL

We show empirically how effective the components presented in Section 2 are by performing five ablations. Each ablation is assessed by the performance on five Atari games. The results are illustrated in Figure 4. Numerical results can be found in Table 3. We observe that data augmentation, action stacking, frame stacking, and temporal consistency are crucial:

1. *No augmentations*: omitting image augmentations leads to poor performance in all games.
2. *No action stacking*: stacking only the frames but not the actions decreases the overall performance for all five games.
3. *No frame stacking*: stacking only the actions but not the frames decreases the performance of all games, with complete failures in Boxing and Breakout. However, Kung Fu Master suffers only a small degradation, possibly because all enemies face the direction they are heading, so their velocity is identifiable from a single frame.
4. *No temporal consistency*: setting the coefficient $\eta$ to zero leads to a significant decrease in performance for most games except Kung Fu Master.
5. *Sample-contrastive*: Garrido et al. (2022) show that VICReg can be also seen as a *dimension-contrastive* method, as opposed to *sample-contrastive* methods such as SimCLR (Chen et al., 2020). They also show that VICReg can be converted to a sample-contrastive method by transposing the embedding matrix $\mathbf{Z}$ in Equation (3). Making VICReg sample-contrastive significantly worsens the performance in Breakout, but less so in the other games.

## 5 COMPARISONS

In this section we compare our method with previous world models regarding the results and the methodology. In Appendix B we discuss the relations to other methods.

### 5.1 RESULT COMPARISON

We compare our method with five baselines: the model-free algorithm SPR (Schwarzer et al., 2021a), with updated scores from Agarwal et al. (2021), and the model-based methods EfficientZero (Ye et al., 2021), IRIS (Micheli et al., 2023), and DreamerV3 (Hafner et al., 2023). The mean metric across all games is calculated using human normalized scores (Mnih et al., 2015).

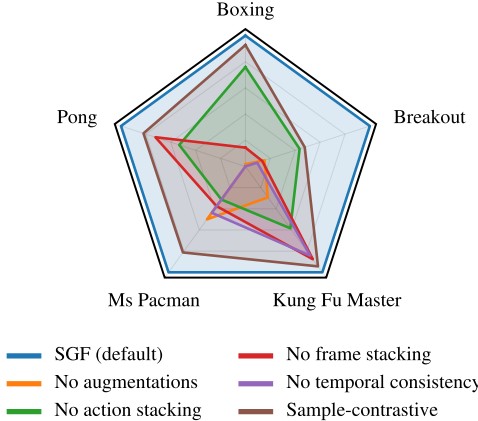

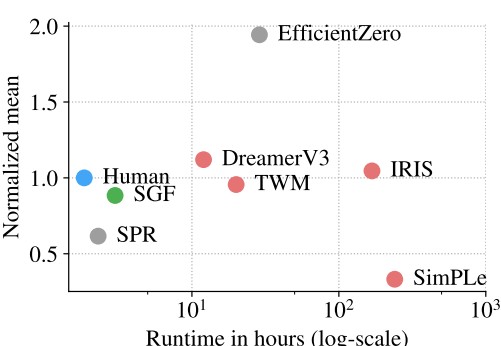

Figure 4: Ablations of SGF in five games. Human normalized scores, normalized with the maximum value achieved per game.

Figure 5: Score and runtime comparison in the Atari 100k benchmark. SPR is model-free, EfficientZero performs lookahead.

In terms of performance, our simple world model achieves good scores with significantly faster training. Figure 5 shows the scores in relation to runtime. Detailed scores can be found in Table 2 and numerical runtimes can be found in Table 4. Training SGF takes 1.5 hours on a single NVIDIA A100 GPU. Obtaining precise training times for other methods is challenging, as they depend on the GPU. Following Hafner et al. (2023), we approximate runtimes for an NVIDIA V100 GPU, assuming NVIDIA P100 GPUs are twice as slow and NVIDIA A100 GPUs are twice as fast. Notably, SGF's runtime is four times shorter than the runtime of DreamerV3, despite both having the same number of imagination steps (1.5 billion). In Table 5 we provide a breakdown of the runtime for certain components of our method.

## 5.2 WORLD MODEL COMPARISON

We also compare the methodologies of SGF and state-of-the-art world models used for learning in imagination: SimPLe (Kaiser et al., 2020), Dreamer (Hafner et al., 2020; 2021; 2023), IRIS (Micheli et al., 2023), TWM (Robine et al., 2023), and the world model developed by Ha & Schmidhuber (2018), referred to as HS. In the Dreamer line of work, our focus is on DreamerV2 and DreamerV3, given their performance improvements over the initial version, notably achieved through the discretization of representations. A summarized comparison of the main differences is provided in Table 1 (we highlight the components in the text).

**Temporal consistency.** We enforce successive representations to be similar by the choice of our objective (*Consistency*). IRIS and HS have no explicit concept of temporal consistency. Dreamer and TWM also seek temporal consistency and attract the representations slightly towards the outputs of the transition predictor from the previous time step. The transition predictor can be interpreted as a time-dependent prior for a variational auto-encoder. An illustration of this difference can be seen in Figure 6. Our approach is simpler for the following reasons:

- In Dreamer, the training of the representation model and the dynamics model is intertwined. Correctly balancing the representation loss and the dynamics loss is crucial for ensuring stable training. Our representation model learns in isolation, simplifying hyperparameter tuning while still achieving temporal consistency.

- In Dreamer, consistency is imposed directly on the representations, whereas we maximize the similarity of the non-linear embeddings of the representations. In DreamerV3 (Hafner et al., 2023) the consistency loss is clipped when it falls below a certain threshold, considering the similarity as sufficient (aka free bits). Our hypothesis is that maximizing the similarity of the embeddings offers a similar degree of freedom.

Table 1: Comparison of methodology with other world models used for imagination. DVx denotes DreamerV2 and DreamerV3. Every component results in additional complexity.

| Component | HS | SimPLe | IRIS | TWM | DVx | SGF |
|---|---|---|---|---|---|---|
| Augmentations | | | | | | x |
| Information Maximization | | | | | | x |
| Stacking | | | | x | | x |
| Consistency | | | | x | x | x |
| Reconstructions | x | x | x | x | x | |
| Discretization | | x | x | x | x | |
| Sequential Dynamics | x | | x | x | x | |
| Stochastic Transitions | x | x | x | x | x | |
| Pixel Transitions | | x | | | | |
| Pixel Dreams | | x | x | | | |
| Act with Memory | x | | x | x | x | |

**Information extraction.**   Previous world models depend on pixel-wise image reconstruction for information extraction from observations (*Reconstructions*). These auto-encoder architectures treat all pixels equally, including less important high-frequency details or noise. In contrast, we adopt a self-supervised objective and utilize data augmentation (*Augmentations*) to learn representations that maximize information (*Information Maximization*) and extract relevant features.

**Discretization.**   Most previous methods learn discrete representations (*Discretization*): SimPLe discretizes representation values into bits. DreamerV2 (Hafner et al., 2021), DreamerV3 (Hafner et al., 2023), and TWM (Robine et al., 2023) utilize softmax normalization to obtain a stack of independent categorical distributions. IRIS (Micheli et al., 2023) converts each image observation into multiple discrete tokens. Discretization introduces additional complexity and requires techniques such as straight-through gradient estimation (Bengio et al., 2013). We propose two primary factors for the success of discretization in world models and explain how they are addressed in our approach:

- Discretization significantly limits the information capacity of representations, potentially preventing collapse in auto-encoder architectures (LeCun, 2022). Since our objective already prevents representation collapse, there is no need for discretization on that account.

- Discretization potentially facilitates dynamics prediction by shrinking and stabilizing the support of $p(\boldsymbol{y}' \mid \boldsymbol{y}, \boldsymbol{a})$. However, we found that a simple architectural choice, specifically adding layer normalization as the final layer of the encoder (as mentioned in Appendix F), is sufficient to keep the mean and variance of the representations stable. Note that another common approach is to normalize the outputs to lie in the interval $[0, 1]$ (Schrittwieser et al., 2020; Schwarzer et al., 2021a).

**Dynamics modeling.**   As we assume Markovian observations, our approach utilizes a feedforward dynamics model. Prior methods, with the exception of SimPLe, are RNN-based (HS, DreamerV2, DreamerV3) or transformer-based (IRIS, TWM) (*Sequential Dynamics*). SimPLe operates directly on image observations instead of operating in a low-dimensional representation space (*Pixel Transitions*). Additionally, our dynamics model is deterministic, whereas previous methods are stochastic (*Stochastic Transitions*), at least regarding transition prediction, while rewards and terminals typically remain deterministic.

**Behavior learning.**   Efficient world models, such as ours, usually train a policy based on the low-dimensional representations. However, IRIS decodes the representations back to pixels, slowing down training significantly (*Pixel Dreams*). Since SimPLe predicts the pixels directly, their policy has to operate on pixels as well. Moreover, previous methods that use a sequence model usually equip the policy with a memory, since the representations encapsulate the history of transitions either via attention or through a compressed recurrent state (*Act with Memory*). Due to our feedforward dynamics model, our policy is memoryless.

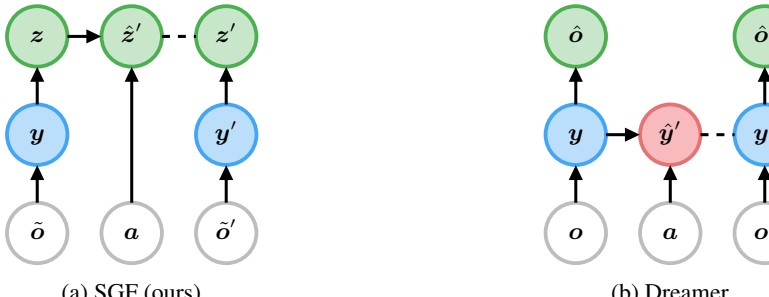

(a) SGF (ours)           (b) Dreamer

Figure 6: High-level illustration of how SGF and Dreamer ensure temporal consistency. The dashed lines denote that two variables are attracted towards each other by a loss function. White nodes indicate inputs, blue nodes indicate representations, and green nodes indicate variables used for representation learning. The red node indicates that Dreamer depends on the dynamics model for representation learning. We omit Dreamer's recurrent states for simplicity.

**Stacking.** Previous world models considered in this section only employ the usual preprocessing, e.g., conversion to grayscale or downscaling the observations. Our method applies frame and action stacking (*Stacking*); TWM applies only frame stacking.

## 6 LIMITATIONS

The insights presented in our paper provide a solid foundation for new world models that are simple, good, and fast. For this paper, we limited our method to deterministic MDPs. To also include non-deterministic POMPDs, the transition distribution needs to be stochastic, allowing for the prediction of multiple possible outcomes. The predictor must also be stochastic to account for non-deterministic information between $o$ and $o'$. Both networks would need to make stochastic predictions, e.g., by modeling the mean and variance of independent normal distributions or using Gaussian mixtures.

Having avoided sequence models such as RNN and transformers, we did limit ourselves to environments with mainly short-term dependencies, which might be the reason that we did not reach state-of-the-art performance. Future work could replace the MLPs used for the transition, reward, and terminal distributions with a sequence model, requiring some implementation effort and resulting in increased computation time. Since the transition distribution is independent of representation learning, this change would not affect the encoder. We evaluated a preliminary version of this in Appendix E. Another more sophisticated approach would involve using a sequence model for the predictor, which would also affect the features extracted by the encoder. However, this would significantly increase the complexity of the model, which we aimed to avoid.

Another current limitation of our approach is that VICReg requires the observations to be images. In principle, VICReg could be applied to other modalities if reasonable augmentations are available. We could also combine SGF with other self-supervised learning methods.

## 7 CONCLUSION

The starting point of our work are the questions: What are the essential components of world models? How far do we get with world models that are not employing RNNs, transformers, discrete representations, and image reconstructions? We demonstrate that representations learned in a self-supervised fashion using VICReg combined with a action-conditioned predictor network and applied to stacked observations can learn latent representations without resorting to resource-intensive sequence models. Self-supervised learning, coupled with augmentations and frame and action stacking, proves effective in building a good world model. Applying SGF to the Atari 100k benchmark, we attained good results with significantly reduced training times. We advocate for future research in model-based reinforcement learning to focus on sparingly adding new components and analyzing their necessity under varying circumstances.

ACKNOWLEDGMENTS

This research has been funded/supported by the Federal Ministry of Education and research of Germany and the state of North Rhine-Westphalia as part of the Lamarr Institute for Machine Learning and Artificial Intelligence.

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

## A  ETHICS STATEMENT

The advancements in developing Simple, Good, and Fast (SGF) world models for reinforcement learning can significantly enhance various fields by making advanced techniques more accessible and reducing computational demands. While this democratization can drive innovation in areas like robotics and autonomous systems, it also raises ethical concerns, such as potential misuse in surveillance or autonomous weaponry. Therefore, it is crucial for the research community to address these risks and develop guidelines to ensure responsible use.

## B  RELATIONS TO OTHER METHODS

### B.1  RECONSTRUCTION-FREE MODELS

Previous world models learning in imagination rely on image reconstructions. However, there are other approaches that learn a model without relying on image reconstructions, which we will discuss in this section.

**Value equivalence.**   In the *value equivalence* paradigm, trajectories of the model must achieve the same cumulative rewards as those in the real environment, regardless of whether the produced hidden states correspond to any real environment states or not. These models are used for decision-time planning. For instance, MuZero (Schrittwieser et al., 2020) trains a model with hidden states by predicting the policy, the value function, and the reward. EfficientZero (Ye et al., 2021) extends this objective by introducing a self-supervised consistency loss with a projector and a predictor network, which shares similarities with our temporal consistency loss. However, they predict the next representation before feeding it into the projector, rather than employing an action-conditioned predictor. Additionally, their approach bears more resemblance to SimSiam (Chen & He, 2021), as they utilize the stop-gradient operation and lack explicit information maximization.

**Auxiliary tasks and bisimulation metrics.**   Learning a model as an auxiliary task can improve the representations of the model-free agent. Although these approaches only share a loose connection to generative world models, they typically are reconstruction-free. For instance, SPR (Schwarzer et al., 2021a) learns a transition model to predict the latent states of future time steps using a projector and a predictor network, incorporating data augmentation. Their architecture shares similarities with ours, however, their transition model is convolutional, and they predict the next representation before feeding it into the projector (similar to EfficientZero). Moreover, their methodology is influenced by BYOL (Grill et al., 2020), utilizing a momentum encoder and lacking explicit information maximization. SPI (Schwarzer et al., 2021b) combines SPR with goal-conditioned RL and inverse dynamics modelling, i.e., predicting the action $a_t$ from states $s_t$ and $s_{t+1}$. They pretrain an encoder on unlabelled data, which is later finetuned on task-specific data.

A special type of auxiliary tasks are connected to bisimulation metrics, where "behaviorally similar" states are grouped together (Ferns & Precup, 2014). Similar to our approach, this also amounts to learning a reward model and a (distributional) latent transition model (by minimizing the Wasserstein distance). Prominent works include DeepMDP (Gelada et al., 2019), which still requires reconstructions for good results on Atari, and DBC (Zhang et al., 2021).

### B.2  SELF-SUPERVISED REPRESENTATION LEARNING

Our self-supervised representation learning framework is similar to existing visual representation learning methods. We describe the differences to the most related works. Note that a common difference is our architecture, as we employ layer normalization instead of batch normalization (Ioffe & Szegedy, 2015), SiLU instead of ReLU nonlinearities, and no ResNet-based encoder (He et al., 2016).

**Relation to VICReg** (Bardes et al., 2022)**.**   Our work is greatly inspired by VICReg, which is used to learn representations of (stationary) images. Originally, the same image is augmented and fed into both branches of the Siamese neural network. We augment two successive image observations, so the two branches get different inputs, which are nonetheless related. Furthermore, VICReg maximizes

the similarity between the embeddings $z$ and $z'$ directly, whereas we integrate an action-conditioned predictor network, since the observations $o$ and $o'$ might have a more complicated connection.

**Relation to BYOL** (Grill et al., 2020) **and SimSiam** (Chen & He, 2021). The basis of our self-supervised learning setup is a combination of ideas from VICReg and BYOL. Specifically, we adopt the predictor network concept from BYOL but utilize the regularization terms from VICReg, omitting BYOL's momentum encoder. Additionally, our predictor network is action-conditioned. Similarly, our method is related to SimSiam (Chen & He, 2021), which also employs a predictor network, but we do not need the stop-gradient operation.

From a practical perspective, these methods could likely achieve comparable performance with appropriate hyperparameter tuning. However, our decision to use VICReg was motivated by its conceptual advantages, which we believe make it particularly suitable for our framework: Specifically, VICReg offers two key advantages over BYOL and SimSiam. First, it avoids the need for additional target networks updated through moving averages (as in BYOL). Second, VICReg has a more established theoretical foundation for its loss functions, leveraging variance and covariance regularization to prevent representation collapse. In contrast, BYOL and SimSiam rely on mechanisms like target networks or stop-gradient operations, which are more heuristic in nature.

### B.3 MODEL-FREE DATA AUGMENTATION

Our method is model-based, but there are model-free methods that also use data augmentation. Laskin et al. (2020a) also use data augmentation to increase the sample efficiency, but by augmenting the image observations passed to the model-free agent. Yarats et al. (2021) additionally regularize the value function such that it is invariant to the augmentations.

# C ADDITIONAL ANALYSIS

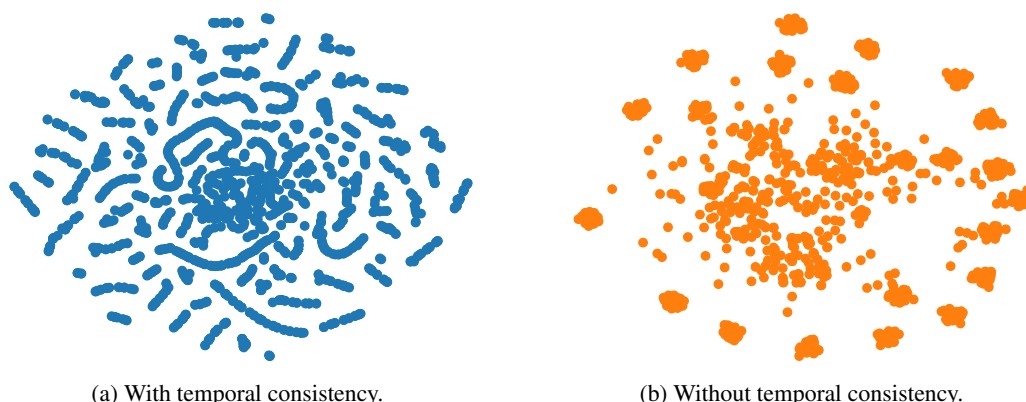

(a) With temporal consistency.

(b) Without temporal consistency.

Figure 7: Comparison of two-dimensional t-SNE embeddings of the learned representations with and without temporal consistency. When only maximizing information, the representations are arranged in Gaussian blobs, which are harder to predict. We show the representations of one episode in Pong.

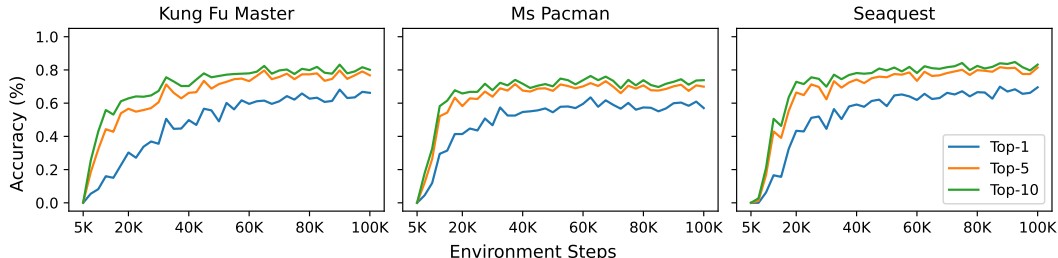

(a) Top-$k$ accuracies of the reconstructions for different values of $k$. The accuracy is determined by first encoding and reconstructing an observation, then computing the MSE between the reconstruction and all ground truth observations in the replay buffer, and finally testing whether the input observation is among the $k$-nearest neighbors. We calculate the mean over a batch of $512$ observations. Note that the observations in the replay buffer can be very similar or even identical, so the top-1 accuracy is not as expressive as top-5 and top-10.

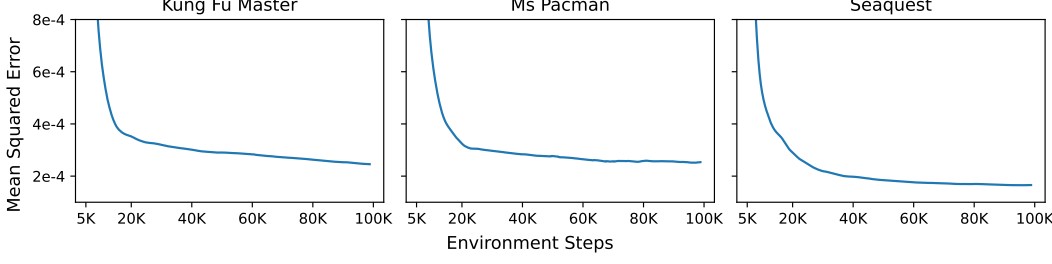

(b) Reconstruction loss of the decoder.

Figure 8: Additional analysis of the decoder from Section 4.1. We start training after collecting 5000 environment steps.

# D DETAILED RESULTS

Table 2: Comparison with other methods on the Atari100k benchmark. Averaged over 10 seeds.

| Game | Random | Human | Model-free SPR | Lookahead Eff. Zero | Learning in imagination IRIS | DreamerV3 | SGF | (std. dev.) |
|------|--------|-------|-----|-----------|------|-----------|-----|-------------|
| Alien | 227.8 | 7127.7 | 841.9 | 808.5 | 420.0 | **959** | 518.8 | (116.7) |
| Amidar | 5.8 | 1719.5 | **179.7** | 148.6 | 143.0 | 139 | 62.7 | (19.4) |
| Assault | 222.4 | 742.0 | 565.6 | 1263.1 | **1524.4** | 706 | 850.1 | (250.3) |
| Asterix | 210.0 | 8503.3 | 962.5 | **25557.8** | 853.6 | 932 | 802.5 | (317.5) |
| Bank Heist | 14.2 | 753.1 | 345.4 | 351.0 | 53.1 | **649** | 58.7 | (38.9) |
| Battle Zone | 2360.0 | 37187.5 | **14834.1** | 13871.2 | 13074.0 | 12250 | 3747.0 | (1240.3) |
| Boxing | 0.1 | 12.1 | 35.7 | 52.7 | 70.1 | 78 | **83.4** | (10.7) |
| Breakout | 1.7 | 30.5 | 19.6 | **414.1** | 83.7 | 31 | 50.7 | (37.6) |
| Chopper Cmd. | 811.0 | 7387.8 | 946.3 | 1117.3 | 1565.0 | 420 | **1775.4** | (593.9) |
| Crazy Climber | 10780.5 | 35829.4 | 36700.5 | 83940.2 | 59324.2 | **97190** | 15751.3 | (5488.9) |
| Demon Attack | 152.1 | 1971.0 | 517.6 | **13003.9** | 2034.4 | 303 | 2809.5 | (749.2) |
| Freeway | 0.0 | 29.6 | 19.3 | 21.8 | **31.1** | 0 | 11.9 | (4.7) |
| Frostbite | 65.2 | 4334.7 | **1170.7** | 296.3 | 259.1 | 909 | 265.6 | (8.6) |
| Gopher | 257.6 | 2412.5 | 660.6 | 3260.3 | 2236.1 | **3730** | 416.4 | (133.7) |
| Hero | 1027.0 | 30826.4 | 5858.6 | 9315.9 | 7037.4 | **11161** | 1522.9 | (1513.1) |
| James Bond | 29.0 | 302.8 | 366.5 | **517.0** | 462.7 | 445 | 280.9 | (60.9) |
| Kangaroo | 52.0 | 3035.0 | 3617.4 | 724.1 | 838.2 | **4098** | 271.2 | (298.0) |
| Krull | 1598.0 | 2665.5 | 3681.6 | 5663.3 | 6616.4 | 7782 | **7813.7** | (1598.0) |
| Kung Fu Master | 258.5 | 22736.3 | 14783.2 | **30944.8** | 21759.8 | 21420 | 20169.8 | (8206.7) |
| Ms Pacman | 307.3 | 6951.6 | 1318.4 | 1281.2 | 999.1 | 1327 | **1356.8** | (775.8) |
| Pong | -20.7 | 14.6 | -5.4 | **20.1** | 14.6 | 18 | 12.6 | (10.0) |
| Private Eye | 24.9 | 69571.3 | 86.0 | 96.7 | 100.0 | **882** | 405.5 | (1144.3) |
| Qbert | 163.9 | 13455.0 | 866.3 | **13781.9** | 745.7 | 3405 | 685.0 | (68.3) |
| Road Runner | 11.5 | 7845.0 | 12213.1 | **17751.3** | 9614.6 | 15565 | 8164.2 | (4066.8) |
| Seaquest | 68.4 | 42054.7 | 558.1 | **1100.2** | 661.3 | 618 | 476.8 | (88.7) |
| Up n' Down | 533.4 | 11693.2 | 10859.2 | **17264.2** | 3546.2 | N/A | 7745.0 | (8515.7) |
| Normalized mean | 0.000 | 1.000 | 0.616 | **1.943** | 1.046 | 1.12 | 0.884 | |
| Normalized median | 0.000 | 1.000 | 0.396 | **1.090** | 0.289 | 0.49 | 0.152 | |

Table 3: Mean scores for the ablation studies.

| Ablation | Boxing | Breakout | Kung Fu Master | Ms Pacman | Pong |
|----------|--------|----------|----------------|-----------|------|
| SGF (default) | 83.4 | 50.7 | 20169.8 | 1356.8 | 12.6 |
| No augmentations | 1.6 | 9.1 | 6031.4 | 847.6 | -20.6 |
| No action stacking | 63.3 | 22.9 | 11981.2 | 651.4 | -3.1 |
| No frame stacking | 12.1 | 8.2 | 17833.2 | 708.4 | 3.2 |
| No temporal consistency | 0.0 | 6.4 | 17008.8 | 754.3 | -20.6 |
| Sample-contrastive | 77.4 | 25.1 | 19237.2 | 1162.6 | 6.4 |

Table 4: Total training times of various methods on the Atari 100k benchmark. They are approximated for an NVIDIA V100 GPU.

| Method | Runtime (hours) |
|--------|-----------------|
| SPR | 2.3 |
| SGF (ours) | 3 |
| DreamerV3 | 12 |
| TWM | 20 |
| EfficientZero | 29 |
| IRIS | 168 |
| SimPLE | 240 |

Table 5: Detailed time breakdown. Percentages in the lower half are relative to the default setting, obtained by enabling or disabling components.

| Component | Percentage |
|-----------|------------|
| Total training | 100 % |
| World model training | 63 % |
| Policy training | 37 % |
| − No augmentations | −16 % |
| − No action stacking | −0.1 % |
| − No frame stacking | −0.1 % |
| + With decoder | +19 % |

# E  ADDITIONAL ABLATIONS

Table 6: Mean scores for additional ablation studies.

| Ablation | Boxing | Breakout | Kung Fu Master | Ms Pacman | Pong |
|---|---|---|---|---|---|
| SGF | 83.9 | 42.2 | 22626.2 | 1134.0 | 14.8 |
| Projector $\times 4$ | 79.1 | 23.7 | 22481.4 | 1176.1 | 6.7 |
| Transition $\times 4$ | 82.9 | 27.1 | 23779.4 | 1208.7 | 13.5 |
| Actor/critic $\times 4$ | 77.9 | 29.7 | 17627.8 | 787.5 | 1.7 |
| Horizon $= 15$ | 57.9 | 30.9 | 21579.4 | 978.1 | -0.4 |
| Training $\times 4$ | 64.1 | 29.0 | 14035.8 | 572.4 | 13.3 |
| Stack size $= 8$ | 68.1 | 21.8 | 19171.2 | 1154.7 | 11.5 |
| Stack size $= 12$ | 35.3 | 14.4 | 16423.6 | 1042.8 | 12.6 |
| Recurrent transition | 71.7 | 11.8 | 18639.4 | 1105.5 | 13 |
| Recurrent predictors | 87.8 | 9.4 | 16086.2 | 1032.1 | 10.5 |

In this section we provide additional ablation studies to analyze the effect of increasing the model size and training time. The results are shown in Table 6. We evaluated these additional ablation studies on 5 instead of 10 random seeds. We observe that the default configuration of SGF performs best in most cases. The ablations are as follows:

1. Projector $\times 4$: We increase the hidden dimension of the projector network from $2048$ to $8192$. This increases the number of parameters of this network from 9.5M to 88M.

2. Transition $\times 4$: We increase the hidden dimension of the transition network from $1024$ to $4096$. This increases the number of parameters of this network from 5.2M to 71.3M.

3. Actor/critic $\times 4$: We increase the hidden dimension of the actor and critic networks from $512$ to $2048$. This increases the total number of parameters of the agent from 0.5M to 8.4M.

4. Horizon $= 15$: We increase the imagination horizon $H$ from $10$ to $15$ steps, and reduce the imagination batch size to 2048 to keep the effective batch size constant.

5. Training $\times 4$: We train the world model and the agent with two batches per environment step instead of one batch every second step. This effectively multiplies the training time by four and is similar to the training time of DreamerV3.

6. Stack size $= 8$: We stack 8 frames and actions instead of $4$.

7. Stack size $= 12$: We stack 12 frames and actions instead of $4$.

8. Recurrent transition: We incorporate a recurrent layer (LSTM) into the transition network, placing it after the five hidden linear layers and before the final output layer. This requires several changes in the implementation, since the recurrent states must be maintained and passed between steps.

9. Recurrent predictors: We make the transition, reward, and terminal distributions recurrent by introducing a shared three-layer MLP followed by an LSTM layer. The output of the LSTM is then fed into a two-layer transition head, as well as the reward and terminal networks.

## F    IMPLEMENTATION DETAILS

Implementing a world model involves numerous design choices, many of which may seem arbitrary at the first glance or are obscured in the source code. In the following, we explain all of our implementation details.

**Stacking and preprocessing.**    As detailed in Section 2.2, we stack the $m$ most recent observations and actions, with $m = 4$. Frame stacking also plays a role in our representation learning approach, with observations $\boldsymbol{o}$ and $\boldsymbol{o}'$ sharing information from three subsequent frames. For data augmentation, we apply the transformations proposed for Atari by Yarats et al. (2021), i.e., random shifts and imagewise intensity jittering.

**Distributions.**    We model the transition distribution using independent normal distributions with unit variance, i.e., $p_\theta(\boldsymbol{y}' \mid \boldsymbol{y}, \boldsymbol{a}) = \mathcal{N}(\boldsymbol{y}' \mid \mu_\theta(\boldsymbol{y}, \boldsymbol{a}), \boldsymbol{I}_d)$, where $\mu_\theta$ is a neural network computing the mean vector. We chose this distribution since the loss function reduces to minimizing the mean squared error $\frac{1}{2}\|\mu_\theta(\boldsymbol{y}, \boldsymbol{a}) - \boldsymbol{y}'\|_2^2$. Also, the mean that we use for prediction is available without further computation. We model the reward distribution $p_\theta(r \mid \boldsymbol{y}, \boldsymbol{a}, \boldsymbol{y}')$ using discrete regression with two-hot encoded targets and symlog predictions, as recently proposed by Hafner et al. (2023). Although not yet being a widely used approach, it makes reward prediction stable across different scales without the need for domain-specific reward normalization or hyperparameter tuning. We model the terminal distribution using a Bernoulli distribution, i.e., $p_\theta(e \mid \boldsymbol{y}, \boldsymbol{a}, \boldsymbol{y}') = \text{Bernoulli}(e \mid \sigma_\theta(\boldsymbol{y}, \boldsymbol{a}, \boldsymbol{y}'))$, where $\sigma_\theta$ is a neural network computing the terminal probability. We chose this distribution since it is a common choice for distributions with binary support; the loss function reduces to the binary cross-entropy, and the mode can be computed by $[\sigma_\theta(\boldsymbol{y}, \boldsymbol{a}, \boldsymbol{y}') \geq 0.5]$ with the squared brackets being Iverson brackets.

**Architecture.**    All networks use SiLU nonlinearities (Hendrycks & Gimpel, 2016) to prevent dead ReLUs, especially given that the data is coming form an ever-changing replay buffer. Furthermore, we employ layer normalization (Ba et al., 2016) in all networks. The encoder $f_\theta$ consists of four convolutional layers with a kernel size of 4, stride of 2, and padding size of 1, followed by a linear layer that computes representations of dimension $d = 512$. To stabilize training, the representations are also normalized using layer normalization; refer to Section 5.2 for our motivation. The projector network is an MLP with two hidden layers of dimension 2048, computing embeddings of dimension $D = 2048$; these dimension have been proven to strike a good balance between qualitative performance and architecture size for VICReg (Garrido et al., 2022). The predictor network uses the same architecture as the projector network. The network of the transition distribution is an MLP with five hidden layers of dimension 1024, and a residual connection from the input to the output. The networks of the reward distribution, terminal distribution, policy, and value function are MLPs with two hidden layers of dimension 1024. We use the AdamW optimizer (Loshchilov & Hutter, 2019) for all networks and loss functions.

**Actor-critic.**    We estimate advantages using generalized advantage estimation (Schulman et al., 2016) and calculate multi-step truncated $\lambda$-returns (Sutton & Barto, 2018) as the target for the value function. To improve exploration and prevent early convergence to suboptimal policies, we add the entropy of the policy to the objective (Williams & Peng, 1991; Mnih et al., 2016). Additionally, we adopt the following strategies from DreamerV3 (Hafner et al., 2023), which have demonstrated success across various environments and reward scales without domain-specific fine-tuning. For advantage computation, the returns are normalized by mapping the 5[th] and the 95[th] percentile to 0 and 1, respectively. The value function utilizes the same discrete regression approach as the reward predictor, i.e., two-hot encoded targets and symlog predictions. A target network, which is the exponential moving average of the online value network, computes additional targets for the value function. This allows for estimating returns using the online network instead of the target network.

---

**Algorithm 1:** SGF's main training procedure.

---

**Input:** environment $\mathcal{E}$, environment steps $M$, imagination horizon $H$
initialize replay buffer $\mathcal{D}$
initialize networks of the world model and the policy
**for** $i \in \{1, \ldots, M\}$ **do**
    execute action in $\mathcal{E}$ according to policy $\pi_\phi$
    store observed transition in $\mathcal{D}$
    # all following computations are batch-wise
    **if** *world model update* **then**
        sample batch of transitions $\tau \sim \mathcal{D}$
        estimate $\mathcal{L}_{\text{Repr.}}(\theta)$ and $\mathcal{L}_{\text{Dyn.}}(\theta)$ according to (2), (4)
        update $\theta$ to minimize the losses
    **if** *policy update* **then**
        # sample from arbitrary time steps
        sample batch of observations $\boldsymbol{O} \sim \mathcal{D}$
        encode observations $\boldsymbol{Y}_1 = f_\theta(\boldsymbol{O})$
        **for** $t \in \{1, \ldots, H\}$ **do**
            select actions $\boldsymbol{A}_t \sim \pi_\phi(\boldsymbol{A}_t \mid \boldsymbol{Y}_t)$
            predict $\boldsymbol{Y}_{t+1} \sim p_\theta(\boldsymbol{Y}_{t+1} \mid \boldsymbol{Y}_t, \boldsymbol{A}_t)$
            predict $\boldsymbol{r}_{t+1} \sim p_\theta(\boldsymbol{r}_{t+1} \mid \boldsymbol{Y}_t, \boldsymbol{A}_t, \boldsymbol{Y}_{t+1})$
            predict $\boldsymbol{e}_{t+1} \sim p_\theta(\boldsymbol{e}_{t+1} \mid \boldsymbol{Y}_t, \boldsymbol{A}_t, \boldsymbol{Y}_{t+1})$
        update $\phi$ actor-critic style using trajectories

---

Table 7: Summary of all hyperparameters. Note that we use the original coefficients for VICReg.

| Hyperparameter | Symbol | Value |
|---|---|---|
| Dimensionality of $y$ | $d$ | 512 |
| Dimensionality of $z$ | $D$ | 2048 |
| Consistency coefficient | $\eta$ | 12.5 |
| Covariance coefficient | $\rho$ | 1.0 |
| Variance coefficient | $\nu$ | 25.0 |
| Frame resolution | – | $64 \times 64$ |
| Grayscale frames | – | No |
| Terminal on loss of life | – | Yes |
| Frame and action stacking | $m$ | 4 |
| Random shifts | – | 0–3 pixels |
| Discount factor | $\gamma$ | 0.997 |
| $\lambda$-return parameter | $\lambda$ | 0.95 |
| Entropy coefficient | – | $1 \times 10^{-3}$ |
| Target network decay | – | 0.98 |
| World model training interval | – | Every 2nd environment step |
| Policy training interval | – | Every 2nd environment step |
| Environment steps | $M$ | 100 000 |
| Initial random steps | – | 5000 |
| World model batch size | – | 1024 |
| World model learning rate | – | $6 \times 10^{-4}$ |
| World model warmup steps | – | 5000 |
| World model weight decay | – | $1 \times 10^{-3}$ |
| World model gradient clipping | – | 10.0 |
| Imagination batch size | – | 3072 |
| Imagination horizon | $H$ | 10 |
| Actor-critic learning rate | – | $2.4 \times 10^{-4}$ |
| Actor-critic gradient clipping | – | 100.0 |
| Policy temperature for evaluation | – | 0.5 (0.01 for Freeway) |
| Random actions during collection | – | 1% |

