# OpenReview forum: "Simple, Good, Fast: Self-Supervised World Models Free of Baggage"
_ICLR.cc/2025/Conference — ICLR 2025 Poster_

### Official Review · Reviewer_iALA · 2024-10-31

**Soundness:** 3
**Presentation:** 3
**Contribution:** 3
**Rating:** 6
**Confidence:** 2

**Summary:**

This paper introduces a world model that uses self-supervised representation learning, captures short-time dependencies through frame and action stacking, and enhances robustness against model errors through data augmentation. This paper is based on a partially observable Markov decision process. It stacks the recent observations and actions to capture short-time dependencies. It introduces stochasticity through data augmentation. To build meaningful representations of observations, this paper enforeces information maximization and temporal consistency of the features.

**Strengths:**

1. This paper contains thorough experiments.
2. The method is simple and fast.

**Weaknesses:**

1. The strategy of stacking observations and actions seem overfitting to the chosen dataset, Atari 100k benchmark, rather than fitting to the real-world, where long-term dependencies are common.
2. Related to the first weakness, while the games in the dataset are deterministic, real-world can be very stochastic. It is questionable whether the model can be applied in real-world cases.

**Questions:**

Considering its simplicity, it would be nice and help the paper strongly if scalability can be demonstrated. Can it be scaled?

---

> ### Author Response · Authors · 2024-11-21
>
> Dear Reviewer,
>
> Thank you for taking the time to review our work and provide your valuable feedback.
> Your comments have been useful in enhancing our paper.
>
> > The strategy of stacking observations and actions seem overfitting to the chosen dataset, Atari 100k benchmark, rather than fitting to the real-world, where long-term dependencies are common.
>
> > Related to the first weakness, while the games in the dataset are deterministic, real-world can be very stochastic. It is questionable whether the model can be applied in real-world cases.
>
> We don't believe that stacking frames and actions is overfitting to Atari, as it is a general and simple method to incorporate short-term information about the dynamics. However, we agree that this approach may not generalize well to real-world environments that require long-term dependencies. Our primary motivation for this work was to provide a simple, yet effective foundation for world models that the research community can build upon. We believe SGF serves as a solid baseline from which future work can explore more complex scenarios and address the specific challenges of real-world environments.
>
> > Considering its simplicity, it would be nice and help the paper strongly if scalability can be demonstrated. Can it be scaled?
>
> We agree with the reviewer's observation regarding the scalability limitations of our approach. However, as is common for deep learning, larger models usually achieve better performance only when accompanied by more data. In this work, we focus on the Atari 100k setting, and the ablation studies in Appendix E demonstrate that extending training time or scaling model size does not yield improvements without more data.
>
> To further address scalability, we included new ablation studies in Appendix E of the revised paper. Achieving higher scores, however, will likely require incorporating additional components, such as stochasticity or long-term sequence modeling, which we propose as directions for future work.

---

> > ### Comment · Reviewer_iALA · 2024-11-24
> >
> > Thanks for the response. However, my concerns remain. I am particularly confused why increasing stack size significantly decreases the performance, as shown in Appendix E. Different from enlarging the network, which the authors argue that the dataset should increase simultaneously to increase performance, increasing stack size does not introduce new parameters, but only more information, which should benefit the performance.

---

> > > ### Author Response · Authors · 2024-11-26
> > >
> > > Thank you for your comment. We agree the performance drop with increased stack size is unexpected. We hypothesize this is due to the higher complexity in the first layer, where filters must process more varied observations simultaneously. This added variability may make it harder for the model to learn meaningful features effectively.
> > >
> > > The purpose of this ablation study was to explore whether simply increasing the stack size could improve performance by capturing more long-term dependencies. However, the results suggest that more sophisticated approaches may be required. For instance, leveraging 3D convolutions or other methods could better utilize the additional context, which we plan to explore in future work.

---

### Official Review · Reviewer_DZ8x · 2024-11-02

**Soundness:** 3
**Presentation:** 2
**Contribution:** 2
**Rating:** 6
**Confidence:** 3

**Summary:**

Authors introduce model-based method: SGF (a Simple, Good, and Fast) world model.
The simplicity of the model is that: it stacks 4 previous frames and actions to capture short-time dependencies, and it uses data augmentation to enhance robustness. It is trained to provide maximum information and temporal consistency.
For simplicity, it does not use: image reconstructions, discretization of representations, sequence models (RNN, transformers, ...),  probabilistic predictions for deterministic environments.

They use several loss functions:
- MSE-loss for temporal consistency (between next embedding and action-conditioned next embedding)
- Variance and Covariance regularization loss for Information Maximization (use the current and next batches of embeddings as input)
- Loss for 3 distributions for Dynamics learning:
    - transition - probability of transition to y'-representation if a-action is applied to y-representation (gradient is only propagated to a-action)
    - reward - probability of r-reward if a-action is applied to y-representation
    - terminal - probability of e-terminal-action if a-action is applied to y-representation

They train several models using AdamW optimizer:
- Encoder (f computes representations from current and next input image observations): 4 convolutional layers (kernel size 4, stride 2, pad 1), linear layer dim = 512, norm-layer, SiLU activation
- Projector (g computes current and next embeddings from current and next representations): MLP with 2 hidden layers of dim = 2048
- Predictor (h predicts action-conditioned next embedding): MLP with 2 hidden layers of dim = 2048
- Transition network: MLP with 5 hidden layers of dim = 1024, and a residual connection from the input to the output (suggested to use the sequence model for future work)
- The networks of the reward distribution, terminal distribution, policy, and value function: MLPs with 2 hidden layers of dim = 1024

This simple approach achieves shorter training times compared to other world models and good performance on the Atari 100k benchmark.

**Strengths:**

In this work, authors try to find the most necessary components that are the most optimal in terms of accuracy and training time.

1. The presented SGF approach lies on the Pareto optimality curve on the chart of Accuracy (normalized mean score) and Training time (hours) - Figure 5, where the other points on the Pareto optimality curve are: SPR, DreamerV3, EfficientZero (performs lookahead)

2. The optimal combination of improvements (frame stacking, action stacking, temporal consistency, augmentations, sample-contrastive) has been found to achieve the highest accuracy in five games - Figure 4

3. Optimal sizes of models and training times have been found in Table 6 to achieve the highest possible mean scores

4. The presented experimental results show the necessity of temporal consistency in Figure 7

**Weaknesses:**

1. While optimal sizes of models and training times have been found in Table 6 to achieve the highest possible mean scores, this may mean that either the scalability of the approach is limited, or the approach must scale in many directions simultaneously to achieve even higher mean scores. Although the approach lies on the Pareto optimality curve on the Accuracy vs Training time chart, i.e. it is one of many optimal options, it is not shown how this approach can be scaled or improved to achieve the highest accuracy with increasing Training time.
Or it requires more serious architectural changes, f.e. as the authors suggest for future work - to use the sequence model for transition network.

2. The proposed method is optimal for tasks that are probably simple enough and do not require remembering very old events, so it is sufficient to have a stack of 4 previous frames and actions, and it is not shown how this approach can be transferred to more complex tasks where orders of magnitude more memory of early events is required.

**Questions:**

Although this is partially explained in the Limitations, could you go into more detail about: for more complex tasks than Atari-games, where longer term knowledge needs to be remembered (where orders of magnitude more memory of early events is required), what parts of the models should be scaled up or significantly modified / replaced (e.g. using  sequence model: RNN, Transformers, ...) to be optimal in terms of accuracy and training time?

---

> ### Author Response · Authors · 2024-11-21
>
> Dear Reviewer,
>
> Thank you for your thoughtful and constructive feedback, as well as the detailed summary.
> We greatly appreciate the time and effort you put into reviewing our work and providing valuable insights.
>
> > While optimal sizes of models and training times have been found in Table 6 to achieve the highest possible mean scores, this may mean that either the scalability of the approach is limited, or the approach must scale in many directions simultaneously to achieve even higher mean scores. Although the approach lies on the Pareto optimality curve on the Accuracy vs Training time chart, i.e. it is one of many optimal options, it is not shown how this approach can be scaled or improved to achieve the highest accuracy with increasing Training time. Or it requires more serious architectural changes, f.e. as the authors suggest for future work - to use the sequence model for transition network.
>
> We agree with the reviewer's observation regarding the scalability limitations of our approach. However, as is often the case in deep learning, larger models tend to achieve better performance only when more data is available. In our study, we focus on the Atari 100k setting, and as shown in the ablation study in Appendix E, increasing training time or using larger models does not yield improvements without additional data. To address scalability further, we have added new ablation studies to Appendix E in our revised paper.
>
> > The proposed method is optimal for tasks that are probably simple enough and do not require remembering very old events, so it is sufficient to have a stack of 4 previous frames and actions, and it is not shown how this approach can be transferred to more complex tasks where orders of magnitude more memory of early events is required.
>
> You are correct that our method performs best in environments where the dynamics do not depend on events that occurred long ago. To address this point further, we have included additional ablation studies in the revised paper (Appendix E), exploring the use of recurrent neural networks in a preliminary manner. The results indicate that incorporating sequence models, at least in their current preliminary form, does not provide a performance benefit in our framework.
>
> The primary objective of this work was to investigate how much performance can be achieved using a minimalist world model with as few components as possible, even on the relatively complex Atari benchmark.
> That said, we recognize the potential value of sequence models and are actively working on integrating them into our world model in a manner that remains as simple and efficient as possible. We believe such work is best suited for a follow-up study to maintain the focus and scope of this paper.
>
> > Although this is partially explained in the Limitations, could you go into more detail about: for more complex tasks than Atari-games, where longer term knowledge needs to be remembered (where orders of magnitude more memory of early events is required), what parts of the models should be scaled up or significantly modified / replaced (e.g. using sequence model: RNN, Transformers, ...) to be optimal in terms of accuracy and training time?
>
> As discussed in our limitations section, there are two natural ways to incorporate sequence models into our framework. The first involves using them for the transition, reward, and terminal distributions. As mentioned above, we added ablation studies in Appendix E of our revised paper, where we describe a preliminary setup: a recurrent layer (LSTM) is added to the transition network, placing it after the five hidden linear layers and before the final output layer. This requires several changes in the implementation, since the recurrent states must be maintained and passed between steps. We also evaluated another variant, as detailed in the updated Appendix E.
>
> The second option is to incorporate sequence models, such as RNNs or Transformers, into the predictor network. However, this would fundamentally change the self-supervised learning signals for the encoder and might impact the training stability of VICReg. Further investigation is needed to ensure robust performance with such modifications. While exploring sequence models is definitely part of our future research plans, we believe their integration is more appropriate for a follow-up study, as it would add additional layers of complexity.

---

> > ### Author Response · Authors · 2024-11-29
> >
> > Thank you again for your valuable feedback. We hope our rebuttal has addressed your concerns and clarified any issues. If there are remaining questions, we would be happy to address them.

---

### Official Review · Reviewer_LRfe · 2024-11-02

**Soundness:** 3
**Presentation:** 3
**Contribution:** 3
**Rating:** 8
**Confidence:** 3

**Summary:**

This paper proposes a new approach to world modeling for the training of RL-based agents. The current state of literature employs different mechanisms to achieve a correct modeling of long sequences, including reconstruction, recurrent modules, memory, etc. The authors propose to simplify the world modeling and just use self-supervised techniques, inspired by VicReg, and a simple learning strategy for the system dynamics. There is a comparison of existing methods and the proposed one on Atari100k, and appropriate ablations of the components.

**Strengths:**

1. I believe these kinds of works are important. It is easy to just incrementally propose new components to improve the performance of systems while considerably increasing the engineering complexity. This does not give a clear view of the actual importance of the components included in the SOTA of world modeling. Going in a completely different direction is in my opinion a needed move sometimes, and it will help to shape new design choices for world modeling. Hence, the motivation is strong, and the reasoning behind that is coherent.
2. The results are compelling. I believe that on Atari100K the simple method proposed performs quantitatively well, even compared to baselines that are far more complex, and with considerably lower runtime.
3. The comparison with existing methods is nontrivial and requires a proper analysis of the literature. Table 1 is also interesting and will be useful for future work.
4. The paper is well-written and well-motivated, all introduced explanations are useful and the writing is compact enough. The proposed experiments are interesting.

**Weaknesses:**

1. I believe that while the proposed method focuses on short-term dependencies, as correctly stated in the limitation, how much performance degrades with an increasingly long-term dependency on actions would be important to quantify. This will allow us to assess the limitations of the proposed method in a more robust manner, for people to build upon.
2. It is not clear to me why only VICReg is chosen for representation extraction. There are relationships with BYOL and SimSiam as reported in the appendix, but it is unclear how performance would change if these approaches were instead used for representation extraction rather than VICReg.

**Questions:**

1. How do the method perform with long-term dependencies?
2. How do feature extraction perform if we change the SSL training objective?

---

> ### Author Response · Authors · 2024-11-21
>
> Dear Reviewer,
>
> Thank you for your thoughtful feedback and for recognizing the contributions of our work.
> We also appreciate the critical points you raised, which have helped us refine and strengthen our paper.
>
> > I believe that while the proposed method focuses on short-term dependencies, as correctly stated in the limitation, how much performance degrades with an increasingly long-term dependency on actions would be important to quantify. This will allow us to assess the limitations of the proposed method in a more robust manner, for people to build upon.
>
> Your point aligns with our broader vision for this research. Indeed, incorporating sequence models is a highly interesting direction that we are actively exploring.
>
> However, the primary motivation of our work was to investigate how far we can get by removing components that increase complexity, both in terms of implementation and runtime. While sequence models are on our research agenda, we believe their integration is best suited for a follow-up study, as they would introduce additional layers of complexity.
>
> To address this, we have conducted new ablation studies (included in Appendix E of our revised paper) that evaluate a preliminary version of this approach using recurrent neural networks. The results indicate that this approach does not lead to performance improvements in our setup. Additionally, we are currently running an experiment with a stack size of 12 to compare with the stack sizes of 4 and 8. We plan to include these results in the second revision before the rebuttal period ends.
>
> > It is not clear to me why only VICReg is chosen for representation extraction. There are relationships with BYOL and SimSiam as reported in the appendix, but it is unclear how performance would change if these approaches were instead used for representation extraction rather than VICReg.
>
> From a practical perspective, these methods could likely achieve comparable performance with appropriate hyperparameter tuning. However, our decision to use VICReg was motivated by its conceptual advantages, which we believe make it particularly suitable for our framework.
>
> Specifically, VICReg offers two key advantages over BYOL and SimSiam. First, it avoids the need for additional target networks updated through moving averages (as in BYOL). Second, VICReg has a more established theoretical foundation for its loss functions, leveraging variance and covariance regularization to prevent representation collapse. In contrast, BYOL and SimSiam rely on mechanisms like target networks or stop-gradient operations, which are more heuristic in nature.
>
> We have included these advantages in Appendix B.2 of the revised paper and hope this explanation provides sufficient context for our choice.
>
> > How do the method perform with long-term dependencies?
>
> We have included an ablation study in Appendix E that incorporates a recurrent neural network to preliminarily evaluate this question.
>
> > How do feature extraction perform if we change the SSL training objective?
>
> We expect other SSL objectives, such as BYOL and SimSiam, to achieve similar performance with appropriate fine-tuning. These methods are relatively similar in design and perform comparably on vision benchmarks, and we see no reason to expect them to perform worse on Atari. However, we chose VICReg for the reasons outlined in our response. Additionally, we experimented with a contrastive variant of VICReg, similar to SimCLR, and observed that it performed worse. This is likely because contrastive methods depend on high-quality negative samples, which are challenging to obtain in the relatively small 100k dataset, especially given the high correlation in the collected trajectories.

---

> > ### Comment · Reviewer_LRfe · 2024-11-27
> >
> > Dear authors,
> > thanks for addressing my doubts. I will keep my rating unchanged.

---

### Official Review · Reviewer_hwrw · 2024-11-04

**Soundness:** 3
**Presentation:** 3
**Contribution:** 1
**Rating:** 3
**Confidence:** 3

**Summary:**

This paper discusses the design of a world model - a parametric model that predicts the transitions probabilities, reward functions and terminal state distributions to serve as a simulation environment for reinforcement learning. The paper proposes to simplify some existing elements in recent world-models, such as sequence models (i.e. RNNs, transformers), to focus on simple ingredients (e.g. frame stacking). It instead proposes to keep the maximum information and temporal consistency formulation as the most essential properties of effective world models. The resultant model is dubbed "Simple, Good and Fast World Models" (SGF) which demonstrates somewhat competitive performance against other existing world models in the Atari 100k benchmark.

**Strengths:**

## Presentation
This paper is well-written with thorough discussion on the related works, the design philosophy and the precise formulations of the proposed modeling. The discussions are usually precise and insightful. The important elements in building the proposed world-models, such as the POMDP formulation, the representation learning (including sufficient details, such as image augmentations, temporal consistency, covariance regularization), the dynamics learning (the conditional independence assumption and the resultant factorization) are all presented clearly, leaving the readers with no doubts on the technical components and the underlying reasoning.

In many cases, the discussions have involved clear contrast with prior works, for example in Table 1 where the design choices in a good selection of prior works are presented, and compared with the design choices in the proposed SGF.

## Empirical Studies
The paper is tested on the standard Atari 100k tasks that is standard for this kind of work. The empirical study seems sounds and demonstrates a few interesting properties of this work that may be useful for researchers in this area. In particular
- It has presented through ablation studies in Section 4.2. showing the importance of various design components, such as state (image) augmentations, action/frame stacking, temporal consistency and sample contrastive formulation. Additional details are also included in the appendix, such as in Table 3 in Appendix D.
- The paper has presented detailed comparisons with other methods, as presented in Appendix D Table 2.

**Weaknesses:**

There are a few limitations that seem to limit the contributions of this work.

- As discussed thoroughly in the related work section in the paper, world model in training reinforcement learning agents is not a new idea. In such cases, it is useful to establish that this work is addressing a significant weakness in prior works, without sacrificing other important metrics. In this case, the main motivation of SGF seems to be presenting a simple, fast, yet accurate method to train a good world model. While SGF certainly fit the bill for the first two, it seems to fall short significantly in the third. The result in Table 2 seems to suggest that SGF is much weaker than prior works such as IRIS and DreamerV3, which are all based on learning on imagination and hence are arguably fairly related.

-  The choice to drop sequence models appears to be a fairly significant limiting factor. This was acknowledged clearly in Line 462-464, which lists the decision to drop sequence models as a potential reasons why this work does not reach SOTA performance on Atari 100k. This seems like a fairly trivial observation - certain games in Atari 100k requires long term reasoning, and it is indeed one of the most challenging aspects of world models for RL agents. It should be expected that removing a components specifically designed to address this important challenge will result in inferior performance. It seems unlikely that researchers in this field will learn much more about how to design better world models if they are simply presented with results comparing methods with and without a sequence model - they probably already know that it is going to be much worse for certain tasks that require long term reasoning.

**Questions:**

I think it will be useful if the paper can present accuracy numbers in similar training time, or training time numbers at roughly the same accuracy numbers. That will be very useful in positioning this work against existing methods.

---

> ### Author Response · Authors · 2024-11-21
>
> Dear Reviewer,
>
> We sincerely appreciate your balanced feedback, recognizing the strengths of our work while pointing out areas for improvement.
> Your comments have helped us to improve our paper.
>
> > As discussed thoroughly in the related work section in the paper, world model in training reinforcement learning agents is not a new idea. In such cases, it is useful to establish that this work is addressing a significant weakness in prior works, without sacrificing other important metrics. In this case, the main motivation of SGF seems to be presenting a simple, fast, yet accurate method to train a good world model. While SGF certainly fit the bill for the first two, it seems to fall short significantly in the third. The result in Table 2 seems to suggest that SGF is much weaker than prior works such as IRIS and DreamerV3, which are all based on learning on imagination and hence are arguably fairly related.
>
> A key goal of our work was to investigate how far we could push performance by removing as many components as possible from world models, rather than aiming to achieve state-of-the-art results. While effective world models such as DreamerV3 and IRIS already exist, these approaches typically require significant computational resources and are more complex to implement.
>
> Creating new world models is a challenging task, and our primary motivation for developing SGF was to offer a simpler and faster approach that could serve as a good baseline for the field. We believe this direction holds value, as it allows a broader range of researchers, including those with limited computational resources, to explore and build upon world models in reinforcement learning.
>
> Despite its minimalism, we think that SGF still achieves competitive results. For example, it outperforms the earlier but more complex "SimPLe" world model while operating with a fraction of the complexity. Moreover, as noted in Table 2, SGF achieves over 70% of DreamerV3's performance and more than 80% of IRIS's performance (in terms of normalized mean on Atari 100k). We consider this as a notable result, demonstrating that good performance is achievable even with minimalistic designs.
>
> > The choice to drop sequence models appears to be a fairly significant limiting factor. This was acknowledged clearly in Line 462-464, which lists the decision to drop sequence models as a potential reasons why this work does not reach SOTA performance on Atari 100k. This seems like a fairly trivial observation - certain games in Atari 100k requires long term reasoning, and it is indeed one of the most challenging aspects of world models for RL agents. It should be expected that removing a components specifically designed to address this important challenge will result in inferior performance. It seems unlikely that researchers in this field will learn much more about how to design better world models if they are simply presented with results comparing methods with and without a sequence model - they probably already know that it is going to be much worse for certain tasks that require long term reasoning.
>
> You are correct that this design choice significantly constrains performance in environments requiring long-term prediction, and we explicitly acknowledge this limitation in the paper.
>
> However, it is important to note that the exclusion of sequence models is not the only simplification in our approach. As outlined in Table 1, our framework also omits several other components commonly found in state-of-the-art world models, such as decoders, discrete categorical representations, and stochastic transitions. From our perspective, one of the key contributions of this work is to demonstrate that many of these components can potentially be removed, and simpler, well-understood components can still yield surprisingly good performance.
>
> Our goal with this paper is to position SGF as a lightweight world model that bridges the gap between purely model-free approaches and complex, resource-intensive model-based methods. While we agree that sequence models are critical for certain tasks, we believe that demonstrating the capabilities of such a minimalistic design provides a valuable baseline and a foundation for future work to build upon.
>
> As discussed in our limitations section, there are two natural ways to incorporate sequence models into our framework. The first is to use them for the transition, reward, and terminal distributions. We conducted ablation studies (included in the revised paper), which preliminarily explore adding a recurrent neural network, but currently they perform similarly. The second option is to apply sequence models to the predictor network, which would fundamentally change the self-supervised learning signals for the encoder. However, this would affect the training stability of VICReg, an approach that requires further investigation to ensure robust performance.

---

> > ### Comment · Reviewer_hwrw · 2024-11-26
> >
> > Thank you for the detailed response to my comments.
> >
> > I agree that it is fairly interesting to understand the "minimum" components required to achieve good performance, to which end the paper has done a decent job. I have already reflected that in my grading of the original review.
> >
> > I am still concerned about the paper's contributions, in view of the significant limitations posted by the lack of sequential models, which notably was sufficiently acknowledged in the initial submission and even more so after the revisions. I think the additional preliminary studies are helpful, but it seems that the results are a bit pre-mature.
> >
> > I would still argue that while there are value in finding out what is "minimum", it is usually necessary for a minimum design to also demonstrate that it is "sufficient" for certain type of tasks, otherwise it is easy to become race to the bottom. Minimalism should be a regularization term, and it cannot be meaningful along with a data term. In this particular case, the data term could likely mean demonstrated SOTA performance in Atari and even more challenging tasks, despite the simple design.
> >
> > Hence while the author responses are helpful, they do not seem to provide new information that is necessary to change the basis of my original score. For this reasons I would like to keep the original rating.

---

> > > ### Author Response · Authors · 2024-11-28
> > >
> > > Thank you for your response and for acknowledging the value of investigating minimal designs. We agree that minimalism must balance simplicity with sufficiency. While SGF does not achieve SOTA, we believe it demonstrates sufficiency: outperforming SimPLe, surpassing prior methods on individual Atari games, and achieving over 70% of DreamerV3's normalized mean.
> > >
> > > Minimalism also has practical advantages beyond theoretical interest. By reducing complexity, SGF lowers the barrier to entry for researchers and offers a stepping stone for future innovations. As Reviewer LRfe aptly noted:
> > >
> > > > It is easy to just incrementally propose new components to improve the performance of systems while considerably increasing the engineering complexity. This does not give a clear view of the actual importance of the components included in the SOTA of world modeling. Going in a completely different direction is in my opinion a needed move sometimes, and it will help to shape new design choices for world modeling.
> > >
> > > We understand your concerns about minimalism becoming a "race to the bottom." However, we believe SGF highlights how simpler approaches can still provide meaningful insights and competitive baselines, contributing to the broader understanding of world models.
> > >
> > > Thank you again for your thoughtful feedback.

---

> ### Author Response · Authors · 2024-11-21
>
> > I think it will be useful if the paper can present accuracy numbers in similar training time, or training time numbers at roughly the same accuracy numbers. That will be very useful in positioning this work against existing methods.
>
> Thank your for this suggestion!
> We are currently running another ablation study with the training time equivalent to DreamerV3. We will add this to our second revision before the rebuttal period ends.

---

### Author Response · Authors · 2024-11-21
**Revision**

Dear Reviewers,

Thank you for your thoughtful feedback. We appreciate the time you took to review our work, and your comments have helped us improve the paper.
We have carefully considered your suggestions and made revisions accordingly.

We have highlight the changes in red in the revised paper. Below is a summary of all the updates we have made:
- We included two ablation studies that combine our world model with recurrent neural networks in a preliminary manner. We achieve similar performance in these experiments.
- We added new ablation studies to Appendix E to further explore the scalability of our method. Specifically, we evaluated scaling the actor and critic networks and are currently assessing the impact of increasing the training time by a factor of four, aligning it with DreamerV3's training time. This evaluation will be finalized in the second revision before the rebuttal period ends.
- Additional details explaining our choice of VICReg over other SSL approaches like BYOL and SimSiam were added to Appendix B.2.
- Upon re-examination, we found that the previously reported training time included evaluation time used for debugging. The corrected runtime for our method is 3 hours (instead of 4.5 hours). Figure 5 and Table 4 have been updated accordingly.
- To improve comparability, we reevaluated all previous additional ablation studies in Appendix E using the same random seeds.

---

> ### Author Response · Authors · 2024-11-26
>
> We have finalized the additional ablation studies, and the updated scores are now available in Table 6 of the revised paper.

---

### Meta-Review · Area_Chair_dGAF · 2024-12-18

**Metareview:**

The paper proposes a new approach to world modeling for the training of RL agents. It aims to find the most necessary components that are the important for achieving a good accuracy and training time. In particular, it uses self-supervised representation learning, captures short-time dependencies through frame and action stacking, and enhances robustness against model errors through data augmentation. For simplicity, it does not use: image reconstructions, discretization of representations, sequence models (RNN, transformers, ...), probabilistic predictions for deterministic environments. Instead, it is trained to provide maximum information and temporal consistency.

The paper is well written and the motivation behind this work is convincing, the experiments are thorough and demonstrate a solid performance, despite the simplicity of the proposed approach. However, the paper got very diverging reviews, with 3/4 reviewers being positive. The main remaining point of discussion is the limited architecture of the model, which prevents long-term planning, a task that is important for agents in real-world situations. The reviewers agreed on this limitation, but disagreed on the impact that this limitation should have on the acceptance of this paper.

The AC benefited significantly from this discussion and, after weighing the positive and critical aspects of the work, agrees with the majority of reviewers, that the paper should be accepted. The AC is convinced of the value of the paper even if the proposed method does not cover problems that require long-term dependency, since 1) problems not requiring long-term dependencies are still significant and 2) establishing a reasonable a good and simple baseline without long-term dependency for people to build upon is of value for itself. Therefore, I recommend acceptance of the paper.

Nevertheless, the authors are encouraged to clarify the scope of the paper early and prominently in the paper, stating the focus on problems that do not require long term dependency.

**Additional Comments On Reviewer Discussion:**

The paper got very diverging reviews, with 3/4 reviewers being positive. The main remaining point of discussion is the limited architecture of the model, which prevents long-term planning, a task that is important for agents in real-world situations. The reviewers agreed on this limitation, but disagreed on the impact that this limitation should have on the acceptance of this paper. Especially, the reviewers on both extremes of the rating scale have exchanged intensely, and stated their arguments very clearly. The AC benefited significantly from this discussion and, after weighing the positive and critical aspects of the work, agrees with the majority of reviewers, that the paper should be accepted.

---

### Decision · Program_Chairs · 2025-01-22

Accept (Poster)